## Research Article

mental health; chronic conditions; community; low-income countries; mental distress

**Corresponding author:**
Tila Mainga;
Email: Tila.Mainga@gmail.com

# Prevalence of mental distress in adults with and without a history of tuberculosis in an urban Zambian community

Tila Mainga[1,2] [iD], Ab Schaap[1,3], Nathaniel Scherer[4], Islay Mactaggart[4],
Kwame Shanaube[1], Helen Ayles[1,3], Virginia Bond[1,2] and Robert C. Stewart[5,6] [iD]

[1]Zambart, School of Public Health, University of Zambia, Ridgeway, Zambia; [2]Department of Global Health and Development, Faculty of Public Health and Policy, London School of Hygiene and Tropical Medicine, London, UK; [3]Clinical Research Department, Faculty of Infectious and Tropical Diseases, London School of Hygiene and Tropical Medicine, London, UK; [4]Department of Population Health, Faculty of Infectious and Tropical Diseases, International Centre for Evidence in Disability, London School of Hygiene and Tropical Medicine, London, UK; [5]Division of Psychiatry, University of Edinburgh, Edinburgh, UK and [6]Malawi Epidemiology and Intervention Research Unit (MEIRU), Lilongwe, Malawi

## Abstract

People with tuberculosis (TB) are susceptible to mental distress. Mental distress can be driven by biological and socio-economic factors including poverty. These factors can persist beyond TB treatment completion yet there is minimal evidence about the mental health of TB survivors. A cross-sectional TB prevalence survey of adults was conducted in an urban community in Zambia. Survey participants were administered the five-item Self Reporting Questionnaire (SRQ-5) mental health screening tool to measure mental distress. Associations between primary exposure (history of TB) and other co-variates with mental distress were investigated using logistic regression. Of 3,393 study participants, 120 were TB survivors (3.5%). The overall prevalence of mental distress (SRQ-5 ≥ 4) in the whole study population was 16.9% (95% CI 15.6%–18.1%). Previous TB history was not associated with mental distress (OR 1.20, 95% CI 0.75–1.92, *p*-value 1.66). Mental distress was associated with being female (OR 1.23 95% CI 1.00–1.51), older age (OR 1.71 95% CI 1.09–2.68) and alcohol abuse (OR 1.81 95% CI 1.19–2.76). Our findings show no association between a previous TB history and mental distress. However, approximately one in six people in the study population screened positive for mental distress.

## Impact statement

There is a growing body of knowledge about the relationship between poor mental health and tuberculosis (TB). Most of this work provides evidence of an association between TB and poor mental health driven by biological, social and economic stressors. Work focused on TB sequelae highlights that these stressors are not necessarily abated by TB treatment completion. Therefore, TB survivors may still hold a disproportionate burden of mental distress. For example, TB survivors may experience continued loss of productivity, coupled with compounded health-seeking costs brought about by morbidity associated with post-TB lung disease. It is plausible that the persistent physical, social and economic stressors associated with TB continue to negatively affect the quality of life of TB survivors. However, there is little evidence examining the prevalence of mental distress in TB survivors and consequently, no recommendations for the management and treatment of mental distress in this population if they carry a disproportionate burden of mental distress. This article contributes to the body of knowledge around the longitudinal relationship between TB and mental health in high TB burdened settings. It is important to establish the mental health burden among TB survivors for appropriate health resource distribution, particularly in settings like Zambia that have some of the highest global TB burdens and lowest mental health resources per capita.

## Introduction

Although there is a growing acknowledgement of the relationship between pulmonary tuberculosis (TB) and mental distress, there is limited longitudinal research on this relationship, despite the documented presence of persistent social (Allwood et al., 2019) and biological (Alene et al., 2021) drivers of mental distress post-TB treatment. Mental distress is a broad concept encompassing common mental health disorders including depression and anxiety as well as general psychological distress defined as non-specific symptoms of stress, anxiety and depression (Ridner, 2004), that does not reach criteria for formal psychiatric diagnosis. This article aims





to assess if people with a history of being treated for TB have a higher mental distress burden than people who have never had TB before.

Approximately 90% of people who fall sick with TB are from 30 low- to middle-income countries (LMIC), including Zambia (World Health Organization, 2021b). In 2020, 59,000 people with TB in Zambia fell ill with TB and 14,800 of them died (World Health Organization, 2021b). The start of the Human Immuno-deficiency Virus (HIV) pandemic in the mid-1980s resulted in a proliferation of TB in Zambia, which was reflected in a 313% increase in the TB notification rate from 124 per 100,000 in 1985 to 512 per 100,000 in the early 2000s (Kapata et al., 2011). HIV is one of the leading drivers of the TB epidemic in Zambia and, according to WHO, Zambia has one of the highest estimated numbers of incident TB cases among people living with HIV (World Health Organization, 2020a). In 2019, 47% of TB patients in Zambia were co-infected with HIV and co-infected individuals accounted for 62% TB related deaths that occurred in that year (World Health Organization, 2020a). In 2019, Zambia had an estimated national HIV prevalence of 11.5% among adults aged 15–49 (UNAIDS, 2019).

The WHO highlights that delays in TB diagnosis can result in poor health outcomes for people with TB, lead to catastrophic costs for their families and continued transmission of TB to others in their communities (World Health Organization, 2015). According to a 2014 TB prevalence survey conducted in Zambia, close to half of people with TB are not diagnosed with TB due to poor medical investigations at the health facility (Kapata et al., 2016); this falls below the WHO current standard of quality assurance which is to diagnose at least 70% of people with sputum smear-positive TB and cure at least 85% of them.

An estimated 66 million people have been cured of TB since the year 2000 (Dodd et al., 2021; World Health Organization, 2021b). According to the World Health Organisation (WHO), a person is cured of bacteriologically confirmed TB if they are smear or culture-negative in the last month of treatment and on at least one previous occasion (World Health Organization, 2013). This definition does not necessarily account for the lasting structural and functional implications of TB on an individual's body (Chakaya et al., 2016). For example, people with TB may develop lung damage during the active or post-treatment phase of the disease (Sarkar et al., 2017), and individuals with a history of TB may have an increased risk of respiratory complications including lung scarring, bronchiectasis, chronic pulmonary aspergillosis, airway stenosis and chronic obstructive pulmonary disease (Byrne et al., 2015; Chakaya et al., 2016; Ravimohan et al., 2018).

The health implications of TB can result in chronic conditions and functional limitations. As a result, when these functional limitations interact with environmental barriers and personal factors, many people with TB experience TB-related disability. A 2021 global systematic review and meta-analysis of 131 studies from 49 countries aimed to document the prevalence of TB-related disability (Alene et al., 2021), defining disability as 'any participation restriction or impairment or activity limitation resulting from TB' (Alene et al., 2021). Their findings showed in addition to mental distress, respiratory impairment was one of the most common types of disability-related impairment resulting from TB in low- to middle-income countries, with a prevalence of 61.2% and 56.1%, respectively (Alene et al., 2021). The prevalence of respiratory impairment may decrease over time, as documented by findings from a Malawian cohort of 405 participants, where the prevalence of respiratory impairment decreased from 60.7% at baseline to 30.7%

after 1 year (Meghji et al., 2020). The handful of studies investigating mental health in TB survivors is suggestive of significant mental distress in this population, particularly in individuals with respiratory symptoms. For example, a 2007 Indian study of 436 TB survivors found that participants who had persistent symptoms, including cough, chest pain and fever, had lower mental well-being scores than those who were symptom-free (55 vs. 77 *p*-value <0.001) (Muniyandi et al., 2007).

In addition to lasting health complications, TB survivors also face lasting negative economic stressors. For example, a 2021 Malawian cohort study that followed up 405 people with TB over the course of 12 months, found that there was a 16.1% increase in the proportion of participants living in poverty 1-year post-treatment completion (Meghji et al., 2021). The study found that, post-TB treatment, the median income of participants was lower (US$44.13 vs. US$72.20) and fewer participants were in paid employment (63% vs. 72.4%, *p*-value = 0.006) (Meghji et al., 2021).

To the best of our knowledge, there are no published studies that have specifically looked at the prevalence of mental distress in TB survivors in the Southern African region, a region with one of the highest incidence of TB (World Health Organization, 2021b). Zambia is among the 20 highest TB burdened countries globally when measured by incidence of TB cases (World Health Organization, 2020a). For people with TB in the country, there is a high prevalence of documented social and economic drivers associated with mental distress (Cremers et al., 2016; Chanda and Sichilima, 2018). People with a history of TB in Zambia are potentially at risk of mental distress, due to long-term health complications and long-term socio-economic stressors (Allwood et al., 2019, 2020; Alene et al., 2021), including the high prevalence of TB stigma. For example, 113 of 138 (81.9%) people with TB reported encountering TB stigma in a 2015 mixed methods study conducted in Lusaka Zambia (Cremers et al., 2015). Stigma has negative implications on the investigation and treatment trajectory of people with TB (Chanda and Sichilima, 2018) and adversely affects mental health. In addition to TB stigma, people with TB are likely to experience intersecting stigmas due to the social and biological realities of their condition. For example, the relationship between TB, poverty and mental distress described above implies that people with TB are at higher risk of being in lower socio-economic brackets and developing mental distress. Being poor (Williams, 2009) and mentally ill (Kapungwe et al., 2010; Mwape et al., 2010) can be stigmatised identities. Additionally, people with TB are likely to be living with other stigmatised co-morbidities including HIV (Hargreaves et al., 2018; Biemba et al., 2019; Bond et al., 2019). In this study, we aimed to compare the prevalence of mental distress in individuals with and without a history of TB in an urban community in Zambia.

## Materials and methods

### *Study design*

Data for this cross-sectional study were collected from one of the 12 urban communities participating in the 'Tuberculosis Reduction through expanded ART and TB Screening' (TREATS) project where 4,000 people were sampled. The main aim of TREATS was to measure the impact of a HIV intervention, that included active case finding for TB, on TB incidence and prevalence. The HIV intervention was part of a previous study known as the 'Population Effects of Antiretroviral Therapy to Reduce HIV Transmission' (HPTN 071, PopART), which was conducted in the same 12 communities in Zambia. A description of the full PopART intervention

is detailed in a previous publication (Hayes et al., 2019). TREATS study activities were conducted at a mobile field site. Study activities included TB screening and testing, and administration of a questionnaire. An opportunity to embed a mental health screening tool into the TREATS questionnaire arose due to a TREATS sub-study that was being done in this particular community as part of the TREATS COVID-19-related response. The study activities were conducted from 18 November 2020 to 24 February 2021. Data collection occurred during the COVID-19 pandemic and therefore all field activities followed strict COVID-19 standard operating procedures to protect researchers and participants.

### Study setting

The community in which the study was conducted is a middle to high-density urban suburb in Kabwe district located in the Central Province of Zambia with a total population of approximately 28,000 people. Infrastructure in the community includes a government health facility, schools, police stations, churches, market areas and transport depots (Bwalya et al., 2020). The majority of residents work in the informal sector as casual labourers or vendors trading in goods (Bwalya et al., 2020). The community has an estimated HIV prevalence of 21.9% (Bond et al., 2021) and an estimated TB prevalence of 0.5%–1% (Ayles et al., 2008). Kabwe was founded around a lead and zinc mine in the early 1900s until the mine formally closed in 1994. The economic implications of the mine closure have tapered off over time (Mankapi, 2011), however, the mining waste still adversely affects residents to date as the town has one of highest levels of lead pollution in the world (Yamada et al., 2023).

### Participants

Participant recruitment began with community engagement activities focused on informing community members about the TREATS study. This was followed by random sampling of households within the community. The random sampling was structured according to geographically defined blocks of around 200 households. All households within these randomly selected blocks were visited by a research assistant who enumerated all household members ($n = 9,533$) and invited all those that were eligible ($n = 6,598$) to take part in the TREATS study activities that were being conducted at a mobile field site that was located as close to the sampled block as possible (Klinkenberg et al., 2023). The eligibility criteria included: being resident in the community; 15 years of age or older at the time of enumeration and ability to provide informed consent. Exclusion criteria included being a participant in a TB vaccine trial or any other TB prevention trial. Of the eligible participants, 3,013 did not attend the mobile field site or consent to be part of the study, and 192 were missing at least one of the mental health indicators used to comprise the mental health score and were therefore excluded. The total number of participants was 3,393.

### Variables

All variables were obtained from the TREATS data set. Mental distress was measured with a screening tool, the five-item Self-Reporting Questionnaire (SRQ-5), that was included in the TREATS questionnaire administered to all participants taking part in the study. The SRQ-5 is a shortened version of the 20-item Self-Reporting Questionnaire (SRQ-20) (Beusenberg et al., 1994). The SRQ-20 was developed by the WHO as a screening tool for

common mental disorders in primary health settings in developing countries. It consists of yes/no items that screen for symptoms of depression, anxiety and somatic manifestations of distress (Beusenberg et al., 1994). The SRQ-5 has been validated in Zambia using the Diagnostic and Statistical Manual of Mental Disorder 4th Edition as a gold standard criterion (Chipimo and Fylkesnes, 2013). It includes 5 items from the SRQ-20; weighted scores on each item are summed to give a total score. In the validation study, the SRQ-5 was translated into both Nyanja and Bemba and had test characteristics similar to the SRQ-20 for detection of mental distress, with an area under the Receiver Operator Curve (ROC) of 0.925 (Chipimo and Fylkesnes, 2013). Similarly, in this study, we also translated the SRQ-5 into Nyanja and Bemba, which are the two most spoken languages in the study community. Translations were done by experienced translators who have an extensive track record with translations of epidemiological questionnaires. Additionally, a validated translated Chewa version of the SRQ 20 was used for cross reference of the Nyanja version as the two languages have strong similarities (Williams, 1996; Stewart et al., 2017). The final translation process included a back-translation stage consisting of 15 Nyanja and/or Bemba speakers who had never seen the SRQ tool before.

The main exposure variable was self-reported history of TB. This was ascertained by a yes/no question that asked if a participant had ever been treated for TB before. Other exposure variables in the questionnaire included in this analysis were sociodemographic characteristics including age, gender, education, employment status and relative social economic status. Relative social economic status was measured using a set of social economic status questions. Health-related characteristics included both self-reported HIV status and HIV testing data, and alcohol use. Alcohol use was measured using the Alcohol Use Disorders Identification Test (AUDIT), with scores of 8 and above suggesting harmful alcohol consumption.

### Data collection

The questionnaire was administered by a team of 18 research assistants (RAs). Most of the RAs were trained psychosocial counsellors with experience administering questionnaires from previous epidemiological studies. The RAs were trained by the first author on administration of the mental health items. The training was guided by 'a *package for data collectors and monitors on the administration of the SRQ'*, adapted from a similar context (Stewart et al., 2017). The training included understanding of depression in people with TB, discussions around the conceptual understanding of each of the mental health items and strategies for making participants feel comfortable when asking sensitive questions. Role-playing sessions further allowed RAs to gain familiarity with administering the tool including translation.

### Sample size

A power calculation was conducted. Estimates from each group were based on prevalence survey participation rates from the nine Zambian communities in which the prevalence survey had previously been completed prior to data collection in this study community. We estimated that approximately 5% of participants were likely to have a history of TB. We estimated the prevalence of mental distress in participants with a history of TB would be 27.9%, and in those without a history of TB to be 14%, based on the prevalence of mental distress in previous studies conducted in

Zambia (Chipimo and Fylkesnes, 2009; van den Heuvel et al., 2013). With a total expected sample size of 2,500 participants and assuming 5% of the participants have a history of TB (125 participants with history of TB and 2,375 with no history of TB) and using a significance level alpha of 0.05, we calculated a power of 81% to detect a difference of 10% in prevalence of mental distress between the 2 groups (25% in the group with a history of TB and 15% in the group with no history of TB).

### Statistical methods

Data was analysed using STATA version 16.0. Demographic, socio-economic and health characteristics of the study population were described using descriptive statistics. Only respondents with no missing mental health data for the five questions that comprise the validated SRQ-5 tool were included in the analysis (N = 3,393).

The outcome variable, mental distress, was analysed as a categorical variable. SRQ-5 total scores range from a minimum of 0 to a maximum of 11 points. The analysis categorised individuals with SRQ scores of 4 or above as screening positive for mental distress. The cut-off of 4 was determined by the previous Zambian validation study. In that study, at cut-off of 4, the SRQ-5 had sensitivity of 0.87 and specificity of 0.85 for detecting 'overall mental distress' as defined by the authors based on DSM-IV diagnoses (Chipimo and Fylkesnes, 2013).

Associations between primary exposure (history of TB) and other co-variates with the outcome were investigated using the chi-square test and logistic regression using a random effects model, adjusting for clustering on sample area. In addition to adjusting for age and sex, the model also adjusted for other risk factors including education, employment status and relative social economic status. The health-related risk factors adjusted for in the model were HIV status and alcohol misuse.

Sensitivity analysis repeated the analyses with different cut-off values of SRQ-5, namely a higher cut-off value of 5 (sensitivity of 0.72 and specificity of 0.94 in the validation study) and a lower cut-off value of 3 (test characteristics not available) (Chipimo and Fylkesnes, 2013). Additional sensitivity analysis was conducted comparing total scores of the outcome variable as a continuous variable. The Wilcoxon Rank Sum Test was used to conduct this sensitivity analysis as the data were not normally distributed.

Note that one SRQ-5 item was mistakenly asked as a positive rather than negative question ('*Are you able to play a useful part in life?*' instead of '*Are you unable to play a useful part in life?*') so this item was reverse scored.

### Ethics and funders

Ethical approval for all study procedures was obtained from the institutional ethics committee at the London School of Hygiene and Tropical Medicine (REF: 14985), and the Bio-medical Ethics Committee of the University of Zambia (REF: 005/02/18). Unique ethical considerations pertaining to conducting research during a pandemic included the fulfilment of additional requirements to ensure the safety of study participants and research teams during the pandemic. Written informed consent was obtained from all participants prior to data collection activities. An assent form was used for participants below the age of 18. Additionally, permission for participation was obtained for participants below the age of 18 from their parents and legal guardians through a signed consent form.

### Results

### Participants

There were 9,533 enumerated individuals from the TREATS blocks, of whom 2,935 did not meet the eligibility criteria, leaving a total of 6,598 eligible participants invited to the mobile field site. Of the eligible participants in the selected community, 3,585 attended the mobile field site and consented to the study (response rate of 54.3%). However, 192 were missing at least one of the items on the SRQ-5 and were therefore excluded from the analysis. In total, 3,393 participants were included in the analysis (see Figure 1).

### Socio-demographic characteristics

Two thousand and five (2,005) (59%) participants were female and 1,963 (57.4%) were below the age of 30. Over half of the participants (53.9%) had never been married with 2,690 (79.2%) having attained secondary level education or higher, 1,378 (40%) were unemployed and 392 (11%) were in formal employment.

### History of TB and other health characteristics

One hundred and twenty (120) (3.5%) participants had a history of TB; 5 (1.3%) tested positive for HIV during the study HIV testing and counselling activity and 94 (14.4%) self-reported to be living with HIV. One hundred and thirty-three (133) (3.9%) participants had an AUDIT score of 8 and above suggestive of alcohol misuse.

### Prevalence of mental distress

The overall prevalence of mental distress (SRQ-5 ≥ 4) in the total study population was 16.9% (95% CI 15.6%- 18.1%). The prevalence of mental distress among people with a history of TB was 21.7% (95% CI 14.6%–30.1%) compared to 16.7% (95% CI 15.4%–18.0%) in individuals with no history of TB. This difference was not significant in the unadjusted model (OR 1.38, 95% CI 0.87–2.15, *p*-value 0.166).

In the univariate analysis, mental distress was associated with being female, older, widowed and alcohol misuse. Women had a higher prevalence of mental distress (18.3%) compared to men (14.8%) (OR 1.28, 95% CI 1.06–1.54, *p*-value <0.001). Individuals who had been widowed reported the highest prevalence of mental distress when compared to individuals who had never been married (26.5% vs. 13.4%) (OR 2.34, 95% CI 1.69–3.23, *p*-value <0.001). Additionally, individuals with an AUDIT score of 8 and above had a higher prevalence of mental distress (26.3%) than those with a score of 7 and below (16.5%) (OR 1.81, 95% CI 1.22–2.69, *p*-value <0.005). The prevalence of mental distress of individuals lowest wealth quintile was higher (19.4%) than that of individuals in the highest wealth quantile (16.6%), however, this difference was not statistically significant in the univariate analysis.

The multivariate model adjusted for age, sex and other risk factors including education, employment status, relative social economic status, HIV status and alcohol misuse. TB history was not associated with mental distress (OR 1.20, 95% CI 0.75–1.92, *p*-value 1.66). Age was associated with mental distress with older individuals almost twice as likely to screen positive for mental distress than those in the youngest age category (15–19 years). HIV status was also associated with mental distress, specifically, participants with unknown status had a lower likelihood of being mentally distressed compared to those with a negative test (OR 0.62, 95% CI 0.47–0.80, *p*-value 0.001). Being female, and

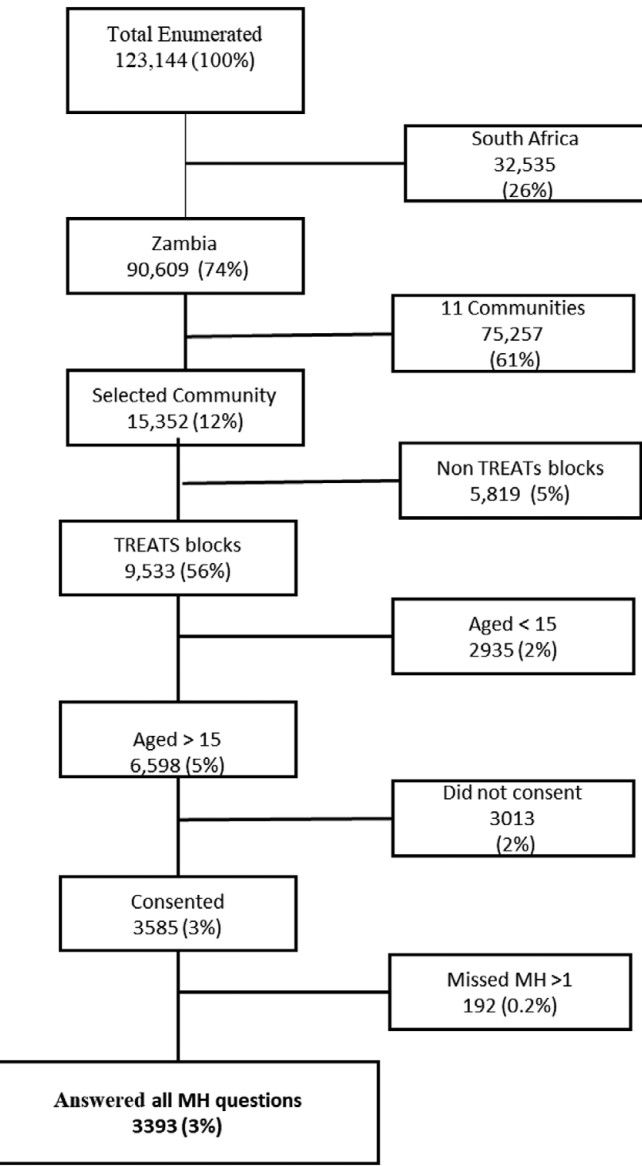

**Figure 1.** Participants flow chart.

alcohol misuse were also associated with mental distress in this model. Table 1 details the univariate and multivariable analysis.

Sensitivity analysis using higher and lower cut-off scores on the SRQ-5 to define mental distress also showed no significant association between a previous history of TB and mental distress (adjusted for age and sex): SRQ-5 cut-off ≥3 (OR 1.11, 95% CI 0.89–1.15, *p*-value 0.82); SRQ ≥ 5 (OR 0.98, 95% CI 0.81–1.78, *p*-value 0.80). Additional, sensitivity analysis using the Wilcoxon Rank Sum Test to compare the difference in means of the total SRQ-5 scores between individuals with a history of TB (mean 2.04 SD 2.3) and those without (mean 1.79 SD 2.2) also found no significant difference (*p*-value 0.23).

## Discussion

To our knowledge, this is the first study in the Southern African region to examine if having a history of TB increases an individual's risk of mental distress. We investigated the prevalence and risk factors of mental distress using a brief validated measure (SRQ-5) and examined associations between mental distress and TB history in our study population. The proportion of participants with a history of TB in our sample was 3.5%, with a 21.7% (95% CI 14.6%–30.1%) prevalence of mental distress. While the prevalence of mental distress in people without a history of TB was 16.7% (95% CI 15.4%–18.0%), there was no statistically significant difference in the prevalence of mental distress between people with a history of TB and those without. Factors associated with mental distress were being female, older, widowed and alcohol misuse.

Finding no difference in the prevalence of distress between people with a history of TB and those without is suggestive that mental distress does not persist beyond TB treatment completion. This finding is corroborated by cohort studies examining trends in the prevalence of mental distress in people with TB over the course of their TB treatment period. For example, a 2016 longitudinal study measuring the prevalence of mental distress in a cohort of 710 people with TB over the 6-month course of their TB treatment

**Table 1.** Univariate and multivariable analyses (*p*-values based on likelihood ratio test)

| Variable | Total cohort (N = 3,393) No. | Total cohort No. % | Mental distress (N = 572) No. | Mental distress No. % | OR (95% CI) | *p*-value | Adjusted OR (95% CI) | *p*-value |
|---|---|---|---|---|---|---|---|---|
| Primary exposure (History of TB) | | | | | | | | |
| No TB history | 3,273 | 96.46 | 546 | 16.7 | 1 | 0.166 | 1 | 0.166 |
| TB history | 120 | 3.54 | 26 | 21.7 | 1.38 (0.87–2.15) | | 1.20 (0.75–1.92) | |
| Social demographics | | | | | | | | |
| Sex | | | | | | 0.009 | | 0.034 |
| Male | 1,388 | 40.9 | 206 | 14.8 | 1 | | 1 | |
| Female | 2,005 | 59.1 | 366 | 18.3 | 1.28 (1.06–1.54) | | 1.23 (1.00–1.51) | |
| Age group | | | | | | <0.001 | | <0.001 |
| 15–19 | 872 | 25.55 | 98 | 11.30 | 1 | | 1 | |
| 20–29 | 1,091 | 32.01 | 164 | 15.10 | 1.39 (1.07–1.82) | | 1.41 (1.01–2.00) | |
| 30–39 | 575 | 16.77 | 122 | 21.44 | 2.14 (1.60–2.86) | | 1.99 (1.32–3.01) | |
| 40–49 | 356 | 10.49 | 78 | 21.91 | 2.20 (1.57–3.05) | | 1.96 (1.24–3.10) | |
| 50+ | 519 | 15.18 | 110 | 21.36 | 2.13 (1.58–2.87) | | 1.71 (1.09–2.68) | |
| Marital status | | | | | | <0.001 | | 0.123 |
| Never married | 1,828 | 53.9 | 244 | 13.4 | 1 | | 1 | |
| Currently married or living as married | 1,139 | 33.6 | 228 | 20.0 | 1.62 (1.33–1.98) | | 1.26 (0.97–1.64) | |
| Divorced or separated | 196 | 5.8 | 39 | 19.9 | 1.61 (1.11–2.35) | | 1.13 (0.73–1.74) | |
| Widowed | 230 | 6.8 | 61 | 26.5 | 2.34 (1.69–3.23) | | 1.72 (1.12–2.64) | |
| Highest level of education completed | | | | | | 0.132 | | 0.762 |
| None | 100 | 3.0 | 24 | 24.0 | 1 | | 1 | |
| Primary school | 603 | 17.8 | 112 | 18.6 | 0.72 (0.46–1.19) | | 0.89 (0.63–1.07) | |
| Secondary school | 2,376 | 70.0 | 382 | 16.1 | 0.61 (0.38–0.97) | | 0.81 (0.72–1.36) | |
| Higher education | 314 | 9.2 | 54 | 17.2 | 0.66 (0.38–1.13) | | 0.78 (0.75–1.49) | |
| Type of employment | | | | | | 0.003 | | 0.635 |
| Unemployed | 1,378 | 40.7 | 247 | 17.9 | 1 | | 1 | |
| Informal employment | 640 | 18.9 | 111 | 17.3 | 0.96 (0.75–1.23) | | 0.82 (0.63–1.07) | |
| Formal employment | 392 | 11.6 | 76 | 19.4 | 1.01 (0.82–1.47) | | 1.00 (0.72–1.36) | |
| Student | 739 | 21.8 | 91 | 12.3 | 0.64 (0.49–0.83) | | 1.06 (0.75–1.49) | |
| Other | 237 | 7.0 | 47 | 19.8 | 1.13 (0.80–1.60) | | 0.89 (0.60–1.28) | |
| Socio-economic position | | | | | | 0.219 | | 0.408 |
| First wealth quantile (poorest) | 670 | 19.8 | 130 | 19.4 | 1 | | 1 | |
| Second wealth quantile | 716 | 21.1 | 123 | 17.2 | 0.86 (0.66–1.13) | | 0.89 (0.67–1.17) | |
| Third wealth quantile | 795 | 23.5 | 117 | 14.7 | 0.72 (0.54–0.94) | | 0.76 (0.58–1.01) | |
| Fourth wealth quantile | 584 | 17.2 | 98 | 16.8 | 0.84 (0.63–1.12) | | 0.87 (0.64–1.17) | |
| Fifth wealth quantile | 625 | 18.4 | 104 | 16.6 | 0.83 (0.62–1.10) | | 0.89 (0.67–1.20) | |

*(Continued)*

**Table 1.** (*Continued*)

| Variable | Total cohort (N = 3,393) No. | Total cohort No. % | Mental distress (N = 572) No. | Mental distress No. % | OR (95% CI) | *p*-value | Adjusted OR (95% CI) | *p*-value |
|---|---|---|---|---|---|---|---|---|
| **Health related characteristics** | | | | | | | | |
| HIV status | | | | | | 0.005 | | 0.001 |
| Negative test | 2,195 | 64.7 | 393 | 17.9 | 1 | | 1 | |
| Positive test | 43 | 1.3 | 5 | 11.6 | 0.60 (0.24–1.56) | | 0.51 (0.20–1.32) | |
| Self-reported positive | 490 | 14.4 | 94 | 19.2 | 1.09 (0.85–1.40) | | 0.77 (0.58–1.02) | |
| Unknown and did not wish to disclose | 665 | 19.6 | 80 | 12.0 | 0.63 (0.48–0.81) | | 0.62 (0.47–0.80) | |
| Alcohol use | | | | | | 0.005 | | 0.031 |
| AUDIT score <7 | 3,260 | 96.1 | 537 | 16.5 | 1 | | 1 | |
| AUDIT Score ≥ 8 | 133 | 3.9 | 35 | 26.3 | 1.81 (1.22–2.69) | | 1.81 (1.19–2.76) | |

found that 34.1% and 81.1% had severe and moderate mental distress respectively at baseline compared to 21.8% and 56.2% after 6 months, representing a reduction of 12.3% and 24.9% in severe and moderate mental distress (Peltzer, 2016). The observed decline in the prevalence of mental distress and depression observed in these studies could be attributed to reduction in both biological and social risk factors of mental distress during the TB treatment period. For example, while on TB treatment, people with TB start to experience less TB morbidity and potentially experience less stigma as they start to look healthier. One of the biggest limitations of our study is that we are unable to explore relationship between time since TB treatment completion and mental distress due to lack of data. However, the above studies suggest that the prevalence of mental distress after TB treatment completion was still relatively high and therefore work focused on understanding the mental distress in TB survivors should also consider the duration post TB treatment as such evidence would provide more comprehensive guidance for mental health and TB treatment.

Our findings suggest that the prevalence of mental distress (16.9%) in the general population in the study setting may be higher than the global of estimate from WHO of approximately 13% (World Health Organization, 2021a). Although our prevalence estimate is based on a cut-off score on a brief mental health screening tool rather than diagnostic interviews, the SRQ-5 is validated in Zambia, and we have presented the test characteristics at the chosen cut-off in this article (Chipimo and Fylkesnes, 2013).

There is some evidence from previous studies that suggests the burden of mental distress in the general Zambian population is high. For example, a 2009 study of 4,466 individuals aged 15–59 established the prevalence of mental distress in the general population in Zambia at 14% (Chipimo and Fylkesnes, 2009). Our findings highlight that mental distress prevalence has remained high since this 2009 study, despite there being a decade gap in which there has been significant funding and innovation in mental health within LMICs. For example, Zambia is among the WHO members that adopted the 2013–2030 WHO comprehensive mental health action plan and an accompanying commitment to meet 10 global targets for improved mental health (World Health Organization, 2021a, 2020b). Other notable innovations include the development of the WHO Mental Health Gap Action

Programme (mhGAP) which has been adopted widely as a means to bridge the mental health treatment gap in low-resource settings (World Health Organization, 2010), including in Zambia (Murray et al., 2015).

The high prevalence of mental distress in this study could be in part due to the fact this research was being conducted during one of the first waves of the COVID-19 pandemic in Zambia. The COVID-19 pandemic was associated with higher levels of distress globally, for example, according to the WHO the COVID-19 pandemic resulted in a 25% increase in the prevalence of anxiety and depression globally (World Health Organization, 2022a). This study contributes to the body of literature regarding the impact of COVID-19 both on TB and mental distress.

The high rates of mental distress could also be attributed to socio-economic stressors. Systematic economic stressors in Zambia over the last decade include electricity load shedding (blackouts) resulting in business failures, particularly for small to medium enterprises, droughts in 2016 and 2019, devaluation of the currency (negatively impacting on imports and availability of credit), and more recently the COVID-19 pandemic (Antoniades et al., 2020). Estimates indicate the growth rate of Zambia's Gross Domestic Product reduced by approximately 4.2% in 2020, and this reduction represented the lowest economic growth in the country since 1998 (National Assembly of Zambia, 2021).

In our sample, we did not find that mental distress was associated with relative socio-economic position. This could in part be explained by the relative homogeneity in the income status of individuals in the surveyed community. The literature highlights that there is a complex bi-directional relationship between mental health and socio-economic conditions, for example, findings from a 2010 systematic review of 115 studies conducted in LMICs highlighted that poverty is both a cause of mental health problems and a consequence: poverty causes poor mental health through social stresses, increased negative life events, stigma and trauma; similarly, mental health problems can lead to impoverishment due to loss of productivity and fragmentation of social relationships (Lund et al., 2010). We considered our finding of a high prevalence of mental distress in our sample through a social determinants of mental health framework, which highlights how socio-economic circumstances in which people live can alter their mental health outcomes. The Lancet Commission on global mental health and

sustainable development acknowledges that addressing these determinants could significantly reduce the global mental health burden and calls for action on the UN Sustainable Development Goals, particularly those focused on economic growth and gender equality (Patel et al., 2018).

It is plausible that the high lead pollution in the town where the study was conducted (despite closure of the lead mine in 1994), has had significant adverse implications on residents' health, furthermore, the town potentially bares a significant social cost associated with lead poisoning (Yamada et al., 2023). These additional stressors could result in worse mental health indicators for Kabwe residents compared to the general Zambian population; unfortunately, the Zambian health system does not track lead-related morbidity and mortality making it difficult to understand the scale of lead poisoning on residents in this town (Human Rights Watch, 2019).

As with our findings, the literature also shows that women are more likely to experience a higher prevalence of mental distress as compared to men. For example, a global meta-analysis of 31 studies ($n = 19{,}639$) found that men had a 37% lower odds of developing depression than women (OR 0.63, 95% CI 0.59–0.68) (Abate, 2013). This relationship holds for various reasons in settings such as Zambia. For example, in Zambia, women are more likely to experience a lack of autonomy and control over their lives resulting in lack of economic and empowerment opportunities, constraints in consumption choices and general lack of basic needs (Aidoo and Harpham, 2001). Women-headed households make up a high proportion of chronically poor households in Zambia (Antoniades et al., 2020). Women living in poverty have high rates of depression as shown by a 2021 global systematic review and meta-analysis of 134 studies ($n = 218{,}035$) which found a 37.4% prevalence of depression among poor women (Corcoran et al., 2021). According to a 2022 systematic review consisting of 14 randomised control trials, psychosocial counselling including individual, manualized interventions and cognitive behavioural therapy was shown to be effective at treating depression in poor women (Corcoran et al., 2022). Task-sharing interventions, such as the Friendship Bench, that use psychosocial counselling and cognitive behavioural therapy delivered by non-specialist personnel have been effective in similar settings (Chibanda et al., 2016) and could be adapted for women experiencing distress in Zambia.

Alcohol misuse was associated with mental distress in our analysis. According to the literature alcohol misuse and mental health conditions, particularly depression, are often co-occurring. For example, a 2021 meta-analysis of 17 studies ($n = 382{,}201$) found that individuals with a mental health condition were twice as likely to report alcohol misuse (OR 2.02, 95% CI 1.72–2.36) (Puddephatt et al., 2021). Similarly, in Zambia, alcohol misuse often co-occurs with both mental and physical conditions. In a 2020 study among adults living with HIV, 94.2% of the 146 participants had unhealthy alcohol use and 72% of these had a mental health condition (Kanguya et al., 2020). Alcohol misuse is a public health concern in Zambia; a population-based survey of 1,928 participants found the overall prevalence of alcohol consumption to be 26.3% (Nzala et al., 2011). This was higher among men (43.5%) than women (17%), with close to half of participants who consumed alcohol reporting that they drunk an average of 5 or more standard alcoholic drinks a day (Nzala et al., 2011). Some studies from this setting reveal that alcohol is used as a means of coping with life stressors including poverty and unemployment (Crane et al., 2018; Taylor et al., 2020).

The literature exploring treatment for mental distress and alcohol misuse provides inconclusive evidence about the best treatment practice. For example, a 2018 systematic review of seven studies that aimed to assess the evidence of interventions targeted at co-occurring depression and alcohol misuse concluded that there is a paucity of evidence on the best clinical practice for treating co-occurring depression and alcohol misuse (Hobden et al., 2018). Similarly, a 2020 systematic review of 21 randomised control trials from 15 different low- to-middle-income countries assessing the effectiveness of psychosocial interventions targeting alcohol misuse found inconclusive results regarding the effectiveness of psychosocial interventions on alcohol misuse (Hobden et al., 2018; Preusse et al., 2020). Therefore, more research needs to be done to understand and treat co-occurring mental distress and alcohol abuse in this context.

We argue that the burden of mental distress in Zambia needs urgent attention, with more investment and effort needed to scale up mental health promotion and treatment. The COVID-19 pandemic further highlighted the importance of mental health (World Health Organization, 2022a). A 2020 WHO estimate suggests the Zambian government spend 0.1% of its budget on mental health and 96.7% of this goes to mental health institutions (World Health Organization, 2020b). However, according to a 2022 WHO report on global mental health the majority of mental health needs can be met outside mental health hospitals (World Health Organization, 2022b). This statement is supported by a growing body of evidence, including a 2021 Cochrane review that shows the effectiveness of task-sharing models being implemented in primary health settings and communities, the review highlights that task sharing approach may increase the number of adults who recover from common mental health conditions in low-to-middle-income countries (van Ginneken et al., 2021). A randomised control trial conducted in Zambia also found this approach to be effective in Zambia as well (Murray et al., 2015). In addition to treatment, the WHO also calls for mitigation of risk factors associated with the development of mental health conditions and this can be done by strengthening the understanding of social and structural determinants of mental health and intervening in ways that reduce risks of developing mental health conditions (World Health Organization, 2022b). In a Zambian context, this would include increasing livelihood support to those in need of it.

## Limitations of the study

The study had some limitations including the fact that the mental health data were collected during the COVID-19 pandemic which may have resulted in a higher prevalence of mental distress than would have been the case if data were collected during more normal circumstances. A further significant limitation was missing data with regards to time since TB treatment completion, which did not allow us to explore the relationship between time since TB treatment completion and individuals' mental health profiles. Furthermore, we were unable to explore the mental health profiles of people with a TB and HIV co-morbidity due to limitation in the number of people with the co-morbidity in the sample. Lastly, there is potential for bias as the results are based solely on individuals who decided to go to the mobile field site, consent to the study and respond to all 5 of the mental health questions in the survey.

## Conclusion

Quantifying the prevalence of mental distress post-TB treatment provides valuable insight into comprehensive TB care, both during

and post, TB treatment. We found no significant statistical association between TB history and mental distress. However, we argue that people with TB in Zambia could benefit from mental health support both during and after their TB treatment. The mental health support should focus on managing and treating mental distress and reducing the prevalent drivers of distress such as stigma, and the economic stressors associated with TB in this context. More research needs to be conducted with a larger investigation group allowing for stratification of the main exposure based on time since TB treatment completion, type of previous TB (multi-drug resistant vs. drug sensitive TB), and number of times an individual had TB in the past. The results from this study indicate mental distress is a significant public health problem in this context, with approximately every one in six people screening positive for probable mental distress in the study population. Mental distress was associated with being older, female, widowed and misusing alcohol. It should be noted that the data was collected during the COVID-19 pandemic and therefore is also indicative of mental distress linked to the pandemic. Urgent measures are needed to understand and mitigate drivers of mental distress in this population.

**Open peer review.** To view the open peer review materials for this article, please visit http://doi.org/10.1017/gmh.2023.83.

**Supplementary material.** The supplementary material for this article can be found at https://doi.org/10.1017/gmh.2023.83.

**Data availability statement.** According to the agreement between the study team and the funder, as laid out in the Data Management Plan, data and metadata (data dictionary) necessary to reproduce the (main) analysis presented in this manuscript will be made publicly and freely available through the LSHTM Compass (https://datacompass.lshtm.ac.uk) after acceptance of the article. The DOI will be assigned to the metadata of the dataset and will be cited in the publication.

**Acknowledgements.** We are grateful to all the participants for giving their time and participating willing. We are also grateful to the TREATS project Publication Working Group for their edits.

**Author contribution.** T.M., A.S. and R.C.S. designed the study. A.S. and T.M. were involved in the data curation, investigation and methodology. T.M. and A.S. coded the data. T.M. led the analysis with oversight from A.S. T.M. wrote the manuscript, which was edited by N.S., A.S., I.M., V.B. and R.C.S. T.M., A.S., N.S., V.B., I.M., R.C.S., K.S. and H.A. contributed to interpretation of the findings and commented on the drafted manuscript. K.S., H.A. and V.B. contributed to funding acquisition. All authors read and approved the final manuscript.

**Financial support.** This research is funded by the EDCTP2 programme supported by the European Union (grant number RIA2016S-1632-TREATS). R.C.S. receives funding from the UK Medical Research Council/GCRF grant to the University of Edinburgh MR/S035818/1.

**Competing interest.** The authors have no competing interests to declare.

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
