## [Reviewer Report]

Dear Editor 

On behalf of my co-authors, I wish to submit a manuscript entitled “Prevalence of mental distress in adults with and without a history of tuberculosis in an urban Zambian community” for consideration by the Cambridge Prisms: Global Mental Health journal.

In this manuscript, we explore whether there is a difference in the prevalence of mental distress in individuals with a history of TB and those without in an urban community in Zambia. Our findings reveal that adults with a history of TB do not have a disproportionate burden of mental distress. However, there is a high prevalence of mental distress in the general Zambian population. Risk factors of mental distress include being female, older and abusing alcohol. 

These findings come at a timely juncture as there is an increased global recognition around the lasting negative physical, social and economic implications of a TB episode on an individuals’ life, all of which serve as drivers of mental distress. However, there is sparse literature exploring the mental health of people with a history of TB despite the acknowledgment of the lasting mental health risk factors that this population face, thereby, potentially overlooking an extremely valuable perspective around TB and mental health. 

We believe that this manuscript is appropriate for publication by the Cambridge Prisms: Global Mental Health journal because the research provides insight into the mental health landscape in Zambia, a country with extensive knowledge gaps around the prevalence and risk factors of mental health conditions in the population. Additionally, the research adds to the body of knowledge around the intersection between chronic illness and mental health in resource limited settings. 

We confirm that this work is original and has not been published elsewhere, nor is it under consideration for publication elsewhere. All authors have approved the manuscript for submission. We have no conflicts of interest to disclose. Please address all correspondence concerning this manuscript to me at tila.mainga@gmail.com; tila.mainga1@lshtm.ac.ukThank you for your consideration of this manuscript.

Sincerely,

Tila Mainga

---

## [Reviewer Report]

Dear authors, thank you for a very comprehensive manuscript that I enjoyed reading. I have a few comments.

Page 3, row 28 sequelae (?)

Row 33 “is” missing

Page 5

Link HIV and TB?

Needs to be explained

Treated TB vs treatment resistant TB?

Def of general psychological distress? (subclinical?)

Background needs to be more comprehensive, say more about TB, illness, treatment available in the sites, stigma, … to show burden of disease to infected people

and why you think people with TB could potentially have a higher rate of CMD (as that seems to have been your hypothesis?). This part doesn’t come out clearly yet.

Page 7

Materials and methods

First time ART is mentioned TREATS

This needs to be in the background to allow for better understanding

Row 131 MFS mobile field site (confusion with msf?)

Recruitment strategy unclear to me… sampling strategy? Everyone?

Row 147: enumeration… did you do a full household sampling process?

How did you invite people to the activities, who was invited, same inclusion strategies?

Row 160: validation of the ChiChewa version of the SRQ 5 only? Needs to be mentioned

Results:

Page 14 row 276 – 280:

Older age category odds of distress versus younger age group, please check the numbers

Discussion;

Your description of depression invites to belief depression to be a mental condition “caused” by situational factors such as poverty (page 18), especially in the genderized studies you mention. This is complicated as depression in the biomedical model is not seen as being caused by events or situations (that only works for trauma), rather these can be mitigating factors (socio-ecological diathesis stress model, see Zuckerman, M. (1999). Diathesis-stress models. In M. Zuckerman, Vulnerability to psychopathology: A biosocial model (pp. 3–23). American Psychological Association. https://doi.org/10.1037/10316-001).

It might be good to discuss that in some form to create an outlook and what can be done with your research findings.

How is the global mental health movement working with the distressing cirumstances that people are in? (great that you mention this on page 20)

Conclusion:

Does TB care include any form of mh support (compare recent efforts to include mh support in HIV care)? Should it in your eyes although you found that there is no significant statistical association?

---

## [Reviewer Report]

This is an important article with insights into the relationship between the history of tuberculosis treatment and the prevalence of mental distress. The article is clear, well written and designed.

First comment: National contextual information is given in the discussion. Are there more local context (related to the sampled community) that could help understand the factors affecting mental health and the results?

2- How was the studied community sampled out of the 12 ones that were in the larger study (sampling of the community)? Can you please also clarify the participant’s sampling method (sampling of the participants)? From my understanding, all 28,000 members of the community’s population were enumerated. And all those who were eligible (how many? 4000? 2500?) were invited. Did the final number of participants depend on those who showed up at the “MFS”? How could that affect the results? Could selection bias result from that and wouldn’t that need to be addressed or acknowledged?

3- Finally, the conclusion has a sentence that generalizes the findings to the whole country of Zambia. From my understanding, strictly speaking, the findings applies to the population of the community where the study has been conducted. I think, if it is the case, that the authors can mention the concordance of their results with other national studies.